# Thalamocortical and corticothalamic pathways differentially contribute to goal-directed behaviors in the rat

Fabien Alcaraz[1,2†], Virginie Fresno[1,2], Alain R Marchand[1,2], Eric J Kremer[3], Etienne Coutureau[1,2], Mathieu Wolff[1,2]*

[1]CNRS, INCIA, UMR 5287, Bordeaux, France; [2]Université de Bordeaux, INCIA, UMR 5287, Bordeaux, France; [3]Institut de Génétique Moléculaire de Montpellier, University of Montpellier, CNRS, Montpellier, France

*For correspondence:
mathieu.wolff@u-bordeaux.fr

Present address: [†]Friedrich Miescher Institute for Biomedical Research, Basel, Switzerland

Competing interests: The authors declare that no competing interests exist.

**Abstract** Highly distributed neural circuits are thought to support adaptive decision-making in volatile and complex environments. Notably, the functional interactions between prefrontal and reciprocally connected thalamic nuclei areas may be important when choices are guided by current goal value or action-outcome contingency. We examined the functional involvement of selected thalamocortical and corticothalamic pathways connecting the dorsomedial prefrontal cortex (dmPFC) and the mediodorsal thalamus (MD) in the behaving rat. Using a chemogenetic approach to inhibit projection-defined dmPFC and MD neurons during an instrumental learning task, we show that thalamocortical and corticothalamic pathways differentially support goal attributes. Both pathways participate in adaptation to the current goal value, but only thalamocortical neurons are required to integrate current causal relationships. These data indicate that antiparallel flow of information within thalamocortical circuits may convey qualitatively distinct aspects of adaptive decision-making and highlight the importance of the direction of information flow within neural circuits.
DOI: https://doi.org/10.7554/eLife.32517.001

## Introduction

To reach specific goals in volatile environments, living organisms must integrate current internal motivational states with an up-to-date causal understanding of the relationships between external events (*Rangel et al., 2008*; *Dickinson, 2012*). Such complex cognitive abilities are supported by highly evolved brain structures. Past research points to a central role for the dorsomedial prefrontal cortex (dmPFC) in adaptive decision-making (*Corbit and Balleine, 2003*; *Killcross and Coutureau, 2003*). A circuit-level analysis of the functional role of the PFC also indicates a major role for areas innervating the PFC (*O'Doherty, 2011*). In this respect, the mediodorsal thalamus (MD) appears of special interest due to the extensive reciprocal projections connecting these two areas (*Groenewegen, 1988*; *Gabbott et al., 2005*; *Alcaraz et al., 2016a*).

These anatomical considerations helped to shape a new functional view of the thalamus, wherein its role is not limited to that of a relay (*Sherman, 2005*; *Mitchell, 2015*; *Wolff et al., 2015a*; *Sherman, 2016*). Indeed experimental interventions aimed at the MD produce a vast array of specific cognitive deficits on both rodents and primates (*Corbit et al., 2003*; *Baxter, 2013*; *Parnaudeau et al., 2013*; *Parnaudeau et al., 2015*; *Alcaraz et al., 2016b*; *Chakraborty et al., 2016*), supporting the view that this thalamic area plays an integrative role within thalamocortical circuits (*Schmitt et al., 2017*). Surprisingly, the functional significance of reciprocal projections, which are the hallmark of thalamocortical organization, has not been directly examined in the context of adaptive decision-making. A recent report provided initial evidence that thalamocortical and

**eLife digest** Planning and decision-making rely upon a region of the brain called the prefrontal cortex. But the prefrontal cortex does not act in isolation. Instead, it works together with a number of other brain regions. These include the thalamus, an area long thought to pass information on to the cortex for further processing. But signals also travel in the opposite direction, from the cortex back to the thalamus. Does the cortex-to-thalamus pathway carry the same information as the thalamus-to-cortex pathway?

To find out, Alcaraz et al. blocked each pathway in rats performing a decision-making task. The rats had learned that pressing a lever led to one type of reward, whereas moving a rod led to another. Alcaraz et al. reduced the desirability of one of the rewards by giving the rats free access to it for an hour. Afterwards, the rats opted mainly for the action associated with the reward that had remained desirable. However, blocking either the thalamus-to-cortex or cortex-to-thalamus pathway prevented this preference from emerging. This suggests that an information flow in both directions is necessary to update knowledge about the value of a reward.

In a second experiment, Alcaraz et al. removed the link between one of the actions and its reward. The reward instead appeared at random, irrespective of the rat's own behavior. Control rats responded by focusing their efforts on the action that still delivered a reliable reward, and by performing the other action less often. Blocking the thalamus-to-cortex pathway prevented this response, but blocking the cortex-to-thalamus pathway did not. This suggests that only the former pathway is necessary to re-evaluate the relationship between an action and an outcome.

Two key aspects of goal-directed behavior – recognizing the value of a reward and the link between an action and an outcome – thus depend differently on the thalamus-to-cortex and cortex-to-thalamus pathways. This same principle may also be at work in other neural circuits with bidirectional connections. Understanding such principles may lead to better strategies for treating disorders of brain connectivity, such as schizophrenia.

DOI: https://doi.org/10.7554/eLife.32517.002

corticothalamic pathways recruited by the same behavioral task may support qualitatively distinct aspects of working memory (*Bolkan et al., 2017*). This further underscores the importance of the functional interactions between cortical and thalamic areas in high-order cognition. Gaining clearer insight into the functioning of thalamocortical circuits therefore requires manipulating thalamocortical and corticothalamic pathways separately.

In the present study, we applied a chemogenetic strategy in rats to specifically inhibit projection-defined dmPFC or MD neurons during a classic instrumental task requiring adaptive actions. We found that distinct goal attributes, namely current goal value and current action-outcome contingency, are differentially supported by thalamocortical and corticothalamic pathways.

## Results

To qualify as 'goal-directed', actions must classically fulfill two criteria: dependence on current goal value and on the causal link between the action and its outcome (*Balleine and Dickinson, 1998*). After an initial instrumental learning phase, both can be assessed separately in the same animals. To gain insights into a potential differential contribution from thalamocortical and corticothalamic pathways, we inhibited projections-defined cortical or thalamic cells during initial training on two distinct actions (pressing a lever or pushing a tilt, see methods) and subsequent choice tests conducted under extinction (*Figure 1*).

### Experiment 1: Thalamocortical pathways are necessary to track changes in both goal value and current instrumental contingency

To express an inhibitory DREADD receptor (*Armbruster et al., 2007*) only in dmPFC-projecting MD cells, an adeno-associated virus carrying a floxed hM4Di receptor expression cassette was injected in the MD, while a retrograde CAV-2 vector (*Junyent and Kremer, 2015*) carrying the Cre recombinase was injected in the dmPFC (*Figure 2A*). As a result, only thalamic cells projecting to the dmPFC were infected by both vectors and therefore expressed mCherry and hM4Di. In general mCherry

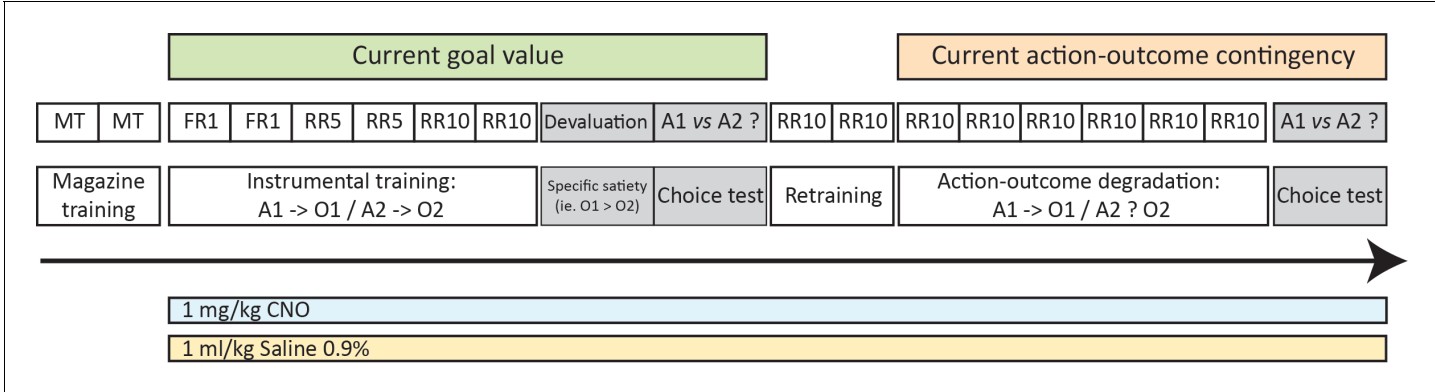

**Figure 1.** Experimental design. After an initial magazine training phase (MT), all rats underwent instrumental training consisting in either pushing a lever or a tilt (see methods) using successively fixed (FR) and random ratios (RR) schedules. To assess how the animals can use current goal value to guide choice, we performed a choice test immediately after selective outcome devaluation, under extinction conditions (both the lever and the tilt are now present in the chamber). After retraining, all rats underwent further instrumental training consisting in a selective degradation procedure (see methods). Another choice test was conducted after this phase, identical to that conducted following outcome devaluation. For both experiments, separate groups of rats were treated with either 1 mg/kg CNO or 1 ml/kg Saline 0.9% given 60 mm prior each behavioral session except during MT. To rule out any potential confounding effect of CNO injection alone, an additional control experiment is provided as an appendix. For each action, instrumental performance during the last RR10 session or the last RR10 retraining session was considered baseline for the devaluation test and the degradation phase, respectively.

DOI: https://doi.org/10.7554/eLife.32517.003

expression was more evident in the lateral portion of the MD, in agreement with our current knowledge of these thalamocortical projections (*Alcaraz et al., 2016a*). mCherry expression was also visible to some degree in adjacent dmPFC-projecting thalamic areas such as the intralaminar group (PC and CL mostly) and, to some extent, the CM and the PV. In some cases, fluorescence was also observed in the habenula. Eight rats showed only minimal (or unilateral) levels of DREADD expression and were therefore excluded from the analyses (saline: n = 7, CNO: n = 7). *Figure 2B and C* illustrate the extent of mCherry expression at the thalamic level.

Instrumental learning took place 1 month postsurgery. Instrumental performance progressively increased over training for both CNO- and saline-treated groups, as shown by the significant effect of Session ($F_{(5,60)} = 11.7$, p<0.0001), but CNO-treated rats tended to perform fewer lever presses overall ($F_{(1,12)} = 11.4$, p=0.0055) (*Figure 3A*). There was no significant Drug X Session interaction however (F < 1), confirming efficient instrumental learning even for the CNO-treated group. In addition, the asymptotical performance did not differ between saline-treated and CNO-treated groups during the final session of training ($F_{(1,12)} = 2.58$, p=0.1338).

During devaluation by specific satiety, both groups of rats consumed the same amount of food (Saline group: 10.8 ± 0.5 g, CNO group: 12.8 ± 0.8 g; $F_{(1,12)} = 1.7$, p=0.2214), indicating that basic motivational processes were not altered by CNO treatment. The ability to use current goal value to guide behavior was assessed during a choice test conducted immediately after devaluation, under extinction conditions. While the group of rats that received saline exhibited the expected adaptive behavior during that test, expressing a clear bias toward the action associated with the still valued outcome, rats that received CNO showed only little differential response toward either actions (*Figure 3B*). Consistent with these observations, the critical Devaluation X Drug interaction approached significance ($F_{(1,12)} = 4.0$, p=0.0679), while the main effect of Devaluation ($F_{(1,12)} = 10.0$, p=0.0081) but not of Drug (F < 1) reached significance. When considering each group separately, a significant effect of Devaluation was evident for the saline ($F_{(1,6)} = 7.7$, p=0.0324) but not the CNO group ($F_{(1,6)} = 2.7$, p=0.1543). To determine if this was due to a performance deficit during this test, we verified the dynamics of responding over time by analyzing the data as 2 min blocks. This analysis confirmed the existence of extinction with a significant effect of Block ($F_{(9,108)} = 8.3$, p<0.0001) and this factor did not interact with Drug (Block X Drug ($F_{(9,108)} = 1.2$, p=0.2908); Block X Devaluation X Drug ($F_{(9,108)} = 1.6$, p=0.1335). Moreover, analyses conducted on each group separately confirmed that responding gradually decreased over time during this test (Saline: Block, F

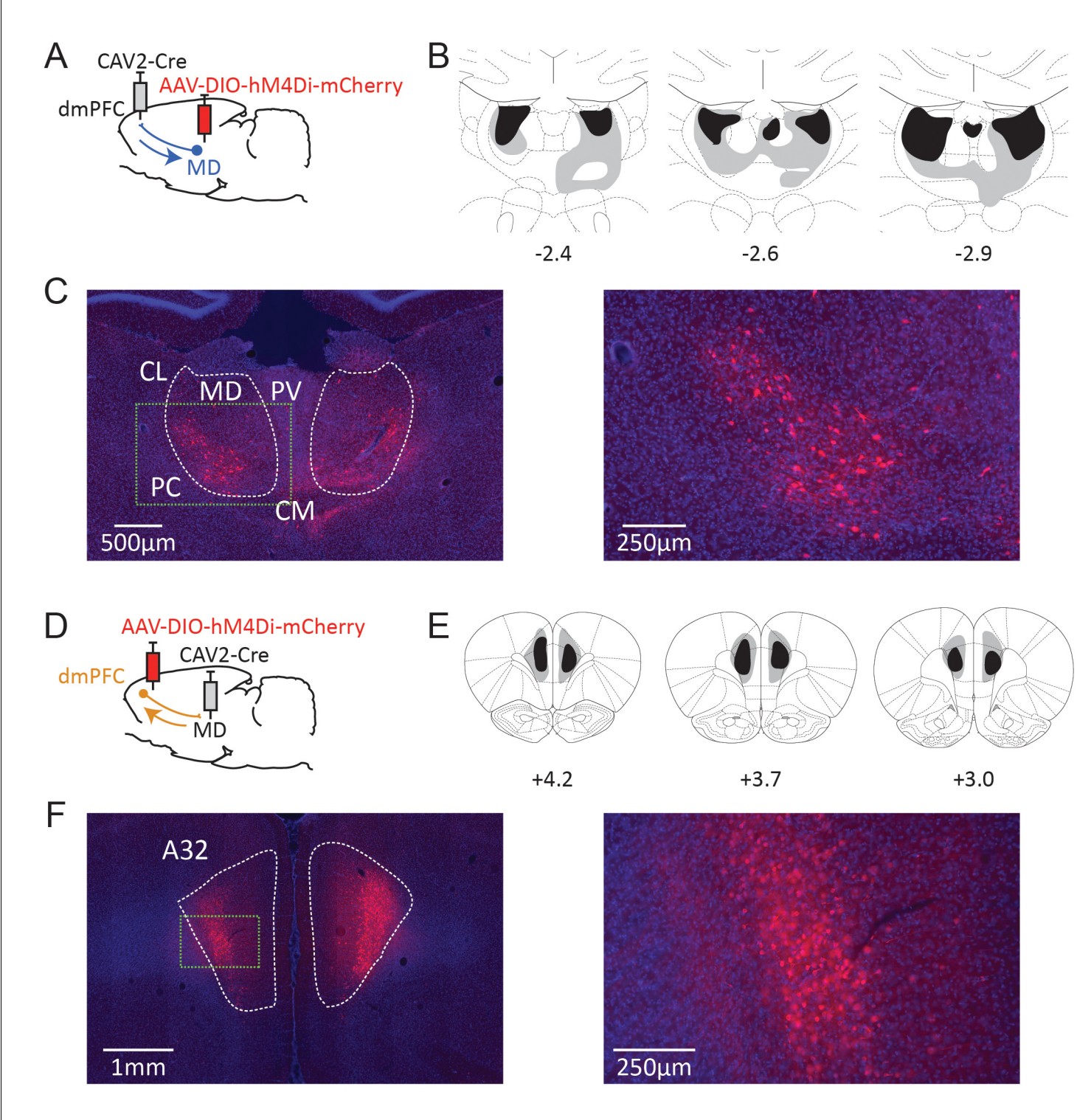

**Figure 2.** Dual-viral chemogenetic strategy to target TC (**A**) and CT pathways (**D**). The CAV-2-Cre vector has a retrograde tropism. Maximal (grey) and minimal (dark) extent of DREADD expression in included rats at three rostrocaudal levels (expressed relative to Bregma, in mm) for CT (**B**) and TC (**E**) pathways. Representative examples of mCherry expression at thalamic (**C**) and cortical (**F**) levels. Insets in dashed green lines correspond to the higher magnification images provided on the right. CL: centrolateral thalamic nucleus, PC: paracentral thalamic nucleus, CM: centromedial thalamic nucleus,

*Figure 2 continued on next page*

*Figure 2 continued*

PV: paraventricular thalamic nucleus. A32 area corresponds to the prelimbic and most dorsal portion of the infralimbic areas in the seventh edition of the Paxinos and Watson atlas (*Paxinos and Watson, 2014*).

DOI: https://doi.org/10.7554/eLife.32517.004

(4,24) = 6.6, p=0.0010; CNO: Block, F(4,24) = 7.4, p=0.0005). Thus, responding was initially higher and then declined to comparable rates for both saline- and CNO-treated groups suggesting that the impairment in the CNO group was not the result of a performance deficit. Collectively, these data therefore suggest that the CNO treatment produced a mild deficit in the ability to update goal value representation and/or its use to guide behavior. Importantly, a consumption test performed immediately after the devaluation test confirmed the effectiveness of the sensory-specific satiety, which was left unaltered by CNO administration. That is, both saline- and CNO-treated rats preferably consumed the still valued outcome when they could freely select from the two outcomes (Devaluation (F (1,12) = 25.6, p=0.0003; Drug and Drug X Devaluation, Fs < 1; *Figure 3C*).

After two sessions of retraining under standard conditions, rats were subjected to a new phase of instrumental training, during which the contingency between one of the actions and its associated outcome was selectively degraded. On this occasion, rats continued to receive the same treatment

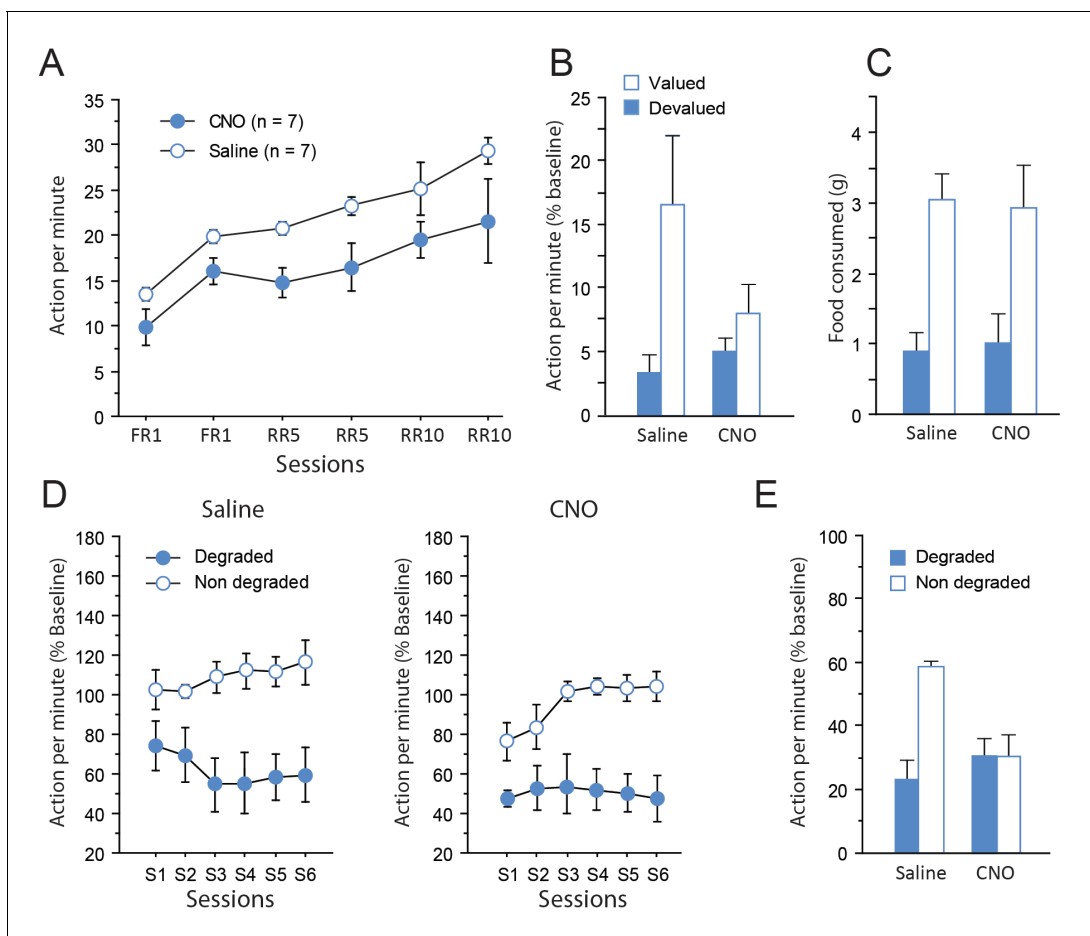

**Figure 3.** Chemogenetic inhibition of TC pathways. (A) Mean number of lever presses (±sem) during instrumental training. (B) Mean number of lever presses (±sem, relative to baseline) during the instrumental choice test conducted immediately after selective outcome devaluation under extinction conditions. (C) Consumption test. (D) Mean number of lever presses (±sem, relative to baseline) during the contingency degradation procedure. (E) Mean number of lever presses (±sem, relative to baseline) for the final choice test conducted under extinction conditions.

DOI: https://doi.org/10.7554/eLife.32517.005

(CNO or saline) as that during initial instrumental learning (see *Figure 1*). During the degradation phase, differential responding was evident for both saline- and CNO-treated groups with lower responding when action-outcome contingency was degraded (*Figure 3D*), as shown by the significant Degradation effect ($F_{(1,12)} = 52.8$, $p<0.0001$) as well as the significant Session X Degradation interaction ($F_{(5,60)} = 3.1$, $p=0.0152$). Drug treatment did not produce any visible effect on this occasion (Drug, $F_{(1,12)} = 1.6$, $p=0.2305$; Session x Drug, $F_{(5,60)} = 1.6$, $p=0.1820$; all remaining Fs < 1). Thus, when the sensory feedback provided by the outcome was available, all rats were capable of exhibiting adaptive decision-making, showing that the consequences of their actions strongly affected their behavior.

Finally, a critical choice test was conducted again under extinction conditions (*Figure 3E)* Interestingly, the behavior exhibited by the two groups of rats was now markedly different. While saline-treated rats continued to express differential responding for both actions, CNO-treated rats were unable to do so. In line with these observations, the critical Degradation X Drug interaction was significant ($F_{(1,12)} = 5.6$, $p=0.0354$), as were the main effects of Drug ($F_{(1,12)} = 6.8$, $p=0.0225$) and of Degradation ($P_{(1,12)} = 6.0$, $p=0.0310$). Further analyses confirmed that the main effect of Degradation was significant for the saline-treated ($F_{(1,6)} = 12.2$, $p=0.0129$) but not the CNO-treated group ($F < 1$).

We provide as an Appendix supplemental data showing that these effect did not result from CNO alone because neither CNO nor DMSO treatment altered behavior throughout testing (*Appendix 1—figures 1–3*). Thus, inhibiting dmPFC-projecting MD neurons produced selective impairments when rats were forced to rely on representations to guide behavior. The impairment appeared to be mild when rats were required to use current goal value, but more pronounced when the contingency between an action and its consequence was altered. Overall, these data indicate a central role for thalamocortical pathways in the ability to guide choice based on current knowledge of the causal link between actions and their outcomes.

## Experiment 2: Corticothalamic pathways are necessary to track changes in goal value but not instrumental contingency

Next, we used the same strategy in a distinct set of rats to examine the behavioral outcome of inhibiting dmPFC neurons projecting to the MD. For this purpose, injections sites for either viral construct were reversed (*Figure 2D*). The resulting mCherry expression at the cortical level is shown in *Figure 2E and F*. A marked expression of mCherry was evident in deep cortical layers, consistent with the existence of abundant corticothalamic projections targeting the MD from cortical layers 5/6 (*Gabbott et al., 2005*). Although we did not quantify the number of labelled cells, comparing experiments 1 and 2 shows that greater labelling was evident for CT cells, consistent with the view that CT cells outnumber TC cells (eg., *Haber and Calzavara, 2009*). Six animals showed little or unilateral DREADD expression and were not considered for analyses (saline: n = 8, CNO: n = 6).

Instrumental learning was comparable between saline- and CNO-treated rats (*Figure 4A*), with improved instrumental learning over training ($F_{(5,12)} = 99.3$, $p<0.0001$). Instrumental learning was not affected by CNO (Drug, $F_{(1,12)} = 3.1$, $p=0.1050$; Session X Drug interaction, $F < 1$).

During devaluation, all rats again consumed an equal amount of food, irrespective of whether they were treated with saline or CNO (Saline group: $9.1 \pm 0.6$ g, CNO group: $9.1 \pm 0.7$ g; $F < 1$). Immediately after the devaluation procedure however, the choice test conducted in extinction revealed a markedly distinct pattern of response in the two groups of rats (*Figure 4B*). While the saline group expressed a clear bias for the still valued option, the CNO group responded similarly for both actions, consistent with the view that they failed to use current goal value to guide behavior. Importantly, the critical Drug X Devaluation interaction reached significance ($F_{(1,12)} = 9.2$, $p=0.0103$), providing compelling support for these observations. In addition, the main effect of Devaluation ($F_{(1,12)} = 8.1$, $p=0.0149$) and Drug ($F_{(1,12)} = 6.6$, $p=0.0244$) also reached significance. Separate analyses confirmed the existence of a selective deficit in CNO-treated (Devaluation, $F < 1$) but not saline-treated (Devaluation, $F_{(1,7)} = 23.1$, $p=0.0020$) rats. Again, the presence of extinction was confirmed by analyzing the data as blocks of 2 min (Block, ($F_{(19,108)} = 3.9$, $p=0.0002$). In addition, drug treatment did not interact with the general dynamics of responding during this test (Block X Drug ($F_{(9,108)} = 1.3$, $p=0.2721$; Block X Devaluation X Drug ($F_{(9,108)} = 1.4$, $p=0.2062$). Further analyses confirmed a significant effect of Block for both saline- ($F_{(4,28)} = 5.0$, $p=0.0035$) and CNO-treated ($F_{(4,28)} = 5.6$, $p=0.0034$) groups suggesting that responding decreased over time in a similar

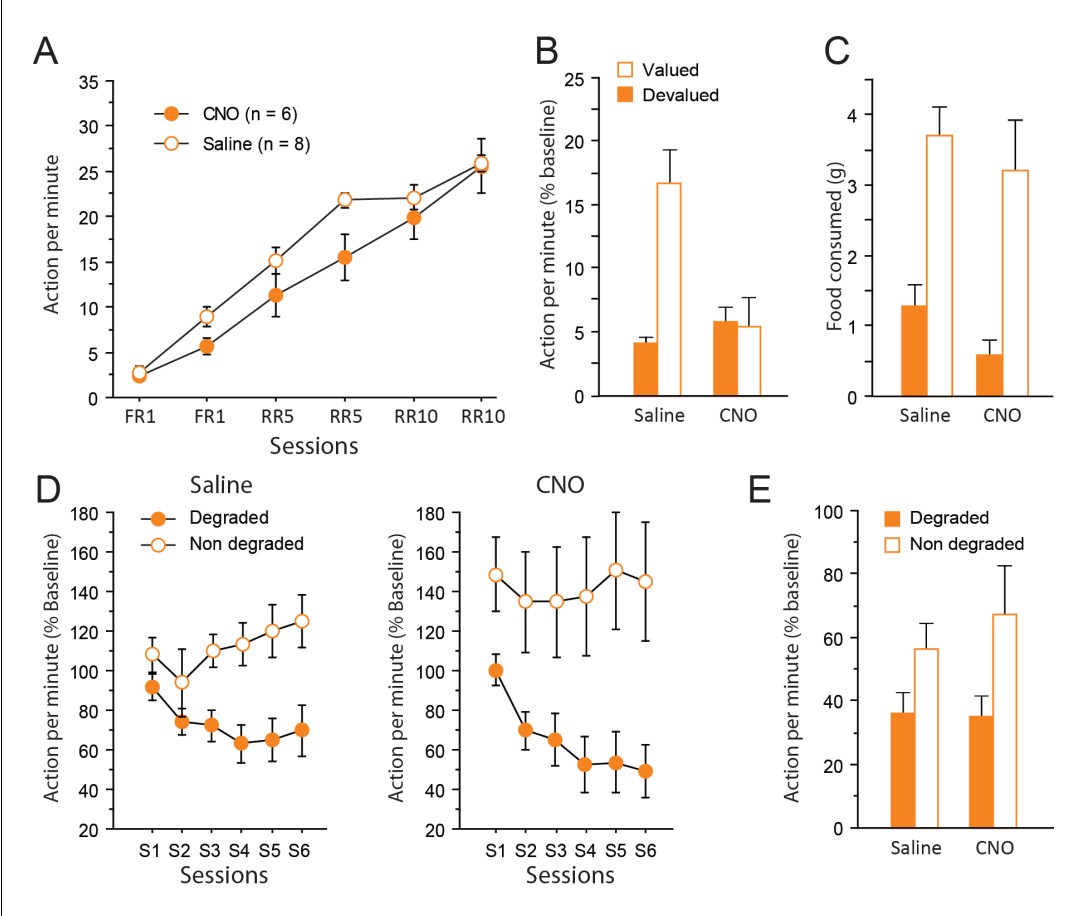

**Figure 4.** Chemogenetic inhibition of corticothalamic pathways. (**A**) Mean number of lever presses (±sem) during instrumental training. (**B**) Mean number of lever presses (±sem, relative to baseline) during the instrumental choice test conducted immediately after selective outcome devaluation under extinction conditions. (**C**) Consumption test. (**D**) Mean number of lever presses (±sem, relative to baseline) during the degradation procedure. (**E**) Mean number of lever presses (±sem, relative to baseline), for the final choice test conducted under extinction.
DOI: https://doi.org/10.7554/eLife.32517.006

fashion for both groups. Thus, floor effect alone cannot account for the specific impairment evident during this test. Consumption tests conducted immediately after yielded essentially the same results as in experiment 1: all rats expressed a clear bias for the still valued outcome (the sensory-specific satiety procedure was efficient) and CNO treatment did not alter behavior at this stage (Devaluation, ($F_{(1,12)}$ = 44.0, p<0.0001; Drug, ($F_{(1,12)}$ = 1.9, p=0.1934; Drug X Devaluation, F < 1, **Figure 4C**).

After two days of retraining, the degradation phase began. Differential responding was evident for both saline- and CNO-treated groups with lower responding when action-outcome contingency was degraded (CNO treatment produced no visible effect, **Figure 4D**). As a consequence, the main effect of Degradation ($F_{(1,12)}$=15.4, p=0.0020) and Session ($F_{(5,60)}$ = 4.6, p=0.0014) as well as the interaction between these factors ($F_{(5,60)}$ = 9.8, p<0.0001) reached significance. The main effect of Drug did not reach significance (F < 1) and no interaction was observed with this factor (Degradation X Drug, $F_{(1,12)}$ = 1.6, p=0.2252; Session X Drug, $F_{(5,60)}$ = 1.6, p=0.1719; Degradation X Session X Drug, F < 1).

During the final choice test conducted in extinction (**Figure 4E**), all rats continued to select the action with reliable consequences as attested by the significant effect of Degradation ($F_{(1,12)}$ = 5.8, p=0.0331). Importantly, inhibiting corticothalamic pathways produced no noticeable effect and did not prevent rats to display adaptive decision-making even when they could only rely on represented information (Drug and Drug X Degradation, Fs <1).

Thus, dissociable patterns of performance were obtained when inhibiting thalamocortical and corticothalamic pathways during choice tests conducted under extinction conditions. A clear deficit in the ability to use recently updated goal value was observed when inhibiting the corticothalamic pathway, but the same treatment did not prevent animals to adapt to a selective change in the contingency between an action and its outcome. The latter ability was however abolished by the inhibition of the thalamocortical pathway, which also produced a mild deficit in the ability to use goal value to guide behavior.

## Discussion

In this study we sought to disentangle the functional contribution of thalamocortical and corticothalamic pathways connecting the dmPFC with the MD in the context of goal-directed behaviors. The present data indicate an important contribution for both cortical and thalamic neurons in the ability to perform adaptive actions. However, while both neuronal populations were found to be important to guide behavior based on current goal value, only thalamic neurons critically supported choice based on current causal relationships. Importantly, inhibiting corticothalamic and thalamocortical pathways produced very specific patterns of behavioral alterations, apparent only when no rewards were available. Thus, deficits only appeared during tests conducted under extinction conditions, suggesting that functional interactions between cortical and thalamic areas are important to guide behavior based on the current content of mental representations. Even then, inhibition of the corticothalamic pathway left the ability to guide choice based on current action-outcome contingency unaltered.

Importantly, control experiments showed that CNO injections alone did not alter instrumental behavior at any stage of the task (Appendix 1). Together with the absence of effects of CNO on consumption (*Figures 3C* and *4C*), this appears to be sufficient to rule out any non-specific impact of CNO, e.g. due to clozapine conversion (*Gomez et al., 2017*). But more importantly, it points to a specific role for projections-defined thalamic and cortical neurons in cognitive processes that are necessary when the task includes unobservable information (*Bradfield et al., 2015*).

Recent studies have emphasized a consistent role for thalamic nuclei to sustain cortical activity when holding online information is important for subsequent choice (*Bolkan et al., 2017*; *Schmitt et al., 2017*). The present data support this view as only dmPFC-projecting MD neurons were found to be important to support choice based on the current mental representation of action-outcome contingency. Thus, one role of the MD could be to provide online information to support choice when no observable element can help to retrieve action-outcome contingency. By itself, this result is also consistent with the effects of global chemogenetic inhibition of the MD (*Parnaudeau et al., 2015*). Similarly, recent findings obtained in primate suggested that MD-lesioned monkeys are unable to persist in successful strategies (*Chakraborty et al., 2016*), hinting at a similar problem of maintaining an accurate representation of the associative structure of the current task over time.

The role of the dmPFC in goal-directed actions is now well established (*Corbit and Balleine, 2003*; *Killcross and Coutureau, 2003*; *Tran-Tu-Yen et al., 2009*; *Hart and Balleine, 2016*), and is crucial for the acquisition of instrumental action-outcome associations (*Tran-Tu-Yen et al., 2009*). However, following acquisition, the role of the dmPFC in adapting instrumental responses to contingency changes appears complex, depending on the accessibility of the reward (*Corbit and Balleine, 2003*) and the nature of contingency changes (*Coutureau et al., 2012*). As a result, the available data therefore suggest that the dmPFC is differentially implicated in guiding choice based on current goal value or current action-outcome contingency (*Naneix et al., 2009*). Our data show that MD-projecting dmPFC neurons were necessary for representing/using current goal value, but not current action-outcome contingency. It is therefore possible that other pathways originating from the dmPFC and preserved in the present study may be relevant to track contingency changes.

Overall levels of responding appeared to be somewhat low during the initial devaluation test, especially when inhibiting dmPFC-projecting MD neurons (Experiment 1). Interestingly, this last feature is reminiscent from classic studies showing lower levels of instrumental performance in rats sustaining MD lesions (*Corbit et al., 2003*). However, in our study the same rats exhibited high level of responding during the degradation phase, suggesting that this disturbance was at best transitory. Low levels of performance are sometimes reported even in controls during devaluation tests

(*Corbit et al., 2003*; *Bradfield and Balleine, 2017*). It seems unlikely that low levels of performance alone could account for the specific impairments during devaluation tests because CNO-treated rats exhibited normal extinction at this occasion. Since responding diminished over time during this test for all rats in a comparable fashion, performance was initially above floor level. We cannot exclude however that the impairments resulted from generalization on the two actions available during the choice test. We were actually concerned beforehand about this possibility, which prompted us to use two clearly distinct manipulanda (a lever and a tilt), unlike the two levers most commonly adopted in the literature. This should limits the possibility that rats generalize current goal value for both actions. In experiment 1 in particular, CNO-treated rats behave as if they were generalizing but this could result from an inability to select the correct option in the absence of the sensory feedback provided by the reward.

The identification of a specific role for corticothalamic projections not only strongly argues against the view of a thalamus acting only as a relay, but also suggests a specific role for these pathways in cognition (*Crandall et al., 2015*; *Guo et al., 2017*). Understanding the functional relevance of these corticothalamic pathways is an important issue as conceptual views posit that they may contribute to cortical functioning by enabling transthalamic communication between cortical areas, thus offering supplemental integrative opportunities (*Sherman and Guillery, 2011*; *Sherman, 2016*). The functional contribution of thalamocortical pathways appears to be consistent with that of a general role of the thalamus to direct attention toward task's elements relevant for successful performance (*Wolff et al., 2015a*; *Wolff et al., 2015b*), not only in the presence of cues, but also when using the current content of mental representation is required for successful performance.

In conclusion, we provide causal evidence that thalamocortical and corticothalamic pathways connecting the dmPFC and the MD support at least partially dissociated goal attributes. These results highlight the directionality of the functional exchanges within neural circuits as one of their fundamental features (see also *Bolkan et al., 2017*; *Lichtenberg et al., 2017*), which calls for a more systematic functional assessment of reciprocally connected pathways. Past research has indicated a time-limited role for both the dmPFC (*Ostlund and Balleine, 2005*; *Tran-Tu-Yen et al., 2009*) and the MD (*Ostlund and Balleine, 2008*) in the acquisition of goal-directed behaviors. While studies that have directly examined functional interactions between cortical and thalamic areas have generally used permanent interventions (*Bradfield et al., 2013*; *Browning et al., 2015*), as was the case in the present study, proceeding to stage-limited interventions appears as a valuable prospect to further refine the functional contribution of projections-defined neurons.

## Materials and methods

### Animals and housing conditions

42 male Long Evans rats weighting 275 g to 300 g at surgery were obtained from Centre d'Elevage Janvier (France). Rats were initially housed in pairs and accustomed to the laboratory facility for two weeks before the beginning of the experiments. Environmental enrichment was provided by tinted polycarbonate tubing elements, in accordance with current French (Council directive 2013–118, February 1, 2013) and European (directive 2010–63, September 22, 2010, European Community) laws and policies regarding animal experiments. The facility was maintained at 21 ± 1°C with lights on from 7 a.m. to 7 p.m. The experimental protocols received approval #5012053-A from the local Ethics Committee on December 7, 2012. After histological verification (see below), the final group sizes were: thalamocortical: n = 7 for saline, n = 7 for CNO; corticothalamic: n = 8 for saline, n = 6 for CNO.

### Surgery

Rats were anaesthetized with 4% Isoflurane and placed in a stereotaxic frame with atraumatic ear bars (Kopf, Tujunga, CA) in a flatskull position. Anaesthesia was maintained with 1.5–2% Isoflurane complemented by subcutaneous administration of buprenorphin (Buprecare, 0.05 mg/kg). CAV-2 and AAV were pressure injected (Picospritzer, General Valve Corporation, Fairfield, NJ) into the brain through a glass micropipette (outside diameter: around 100 μm) and polyethylene tubing. For MD-to-dmPFC pathway targeting, 1 μl of $1 \times 10^9$ genomic copies/μl of CAV2-Cre (Biocampus PVM, Montpellier, France) was injected bilaterally in the PL at the following coordinates: AP +3.2 mm from

bregma, laterality ±0.6 mm, ventrality −3.4 mm from skull. In the same surgery session, 1 µl of 1 × $10^9$ genomic copies/µl of AAV-hSyn-DIO-hM4Di-mCherry (UNC Vector Core, USA) was injected bilaterally in the MD at the following coordinates: AP −2.6 mm, laterality ±0.7 mm and ventrality −5.6 mm. For dmPFC-to-MD pathway targeting, virus injections were reversed, that is, CAV-2 in the MD and AAV in the dmPFC. All injection parameters were the same, except for the mediolateral coordinates of AAV injection in the dmPFC, set at ±0.8 mm, in order to preferentially target the cortical layers V and VI which project to the MD. In all groups, the pipette was left in place 5 min after injection before slow retraction. To allow for optimal viral expression, rats were given one month of recovery before behavioral testing began.

## Behavioral experiments
### Behavioral apparatus
Animals were trained in eight identical conditioning chambers (40 cm wide x 30 cm deep x 35 cm high, Imetronic, France), each located inside a sound and light-attenuating wooden chamber (74 × 46 × 50 cm). Each chamber had a ventilation fan producing a background noise of 55 dB and four LEDs on the ceiling for illumination. Each chamber had two opaque panels on the right and left sides, two clear Perspex walls on the back and front sides and a stainless-steel grid floor (rod diameter: 0.5 cm; inter-rod distance: 1.5 cm). In the middle of the left wall, a magazine (6 × 4.5 × 4.5 cm) received either grain or sucrose pellets (45 mg, F0165, Bio Serv, NJ, USA) from dispensers located outside the operant chamber. The magazine was equipped with infra-red cells to detect the animal's visits. A retractable lever (4 × 1 × 2 cm) could be inserted next to the magazine as did a 'tilt', a vertical rod hinged on the ceiling and terminated by a small plastic ball. Pressing the lever or pushing the tilt in any direction were therefore the two distinct actions that rats could perform during instrumental tasks. Activation of either the lever or the tilt produced the delivery of the associated outcome, as a function of the current procedure (i.e. FR1, RR5 or RR10, see below). A personal computer connected to the operant chambers and equipped with POLY software and interface (Imetronic, France) controlled the equipment and recorded the data.

## Instrumental training
Rats were first habituated to the magazine dispenser through two daily sessions of magazine training for 2 days. A session consisted in the delivery of 30 food rewards, grain or sucrose pellets, distributed randomly through a 30 min session. The first session took place in the morning, and the second in the afternoon, with the order of rewards counterbalanced between rats and days. Twelve daily sessions of instrumental training began the day after the last session of magazine training, during which rats had to make specific associations between two responses (lever press or tilt action) and the two different outcomes. Daily training consisted in instrumental learning with either the lever or the tilt, each specifically associated with one of the outcome (i.e. either grain or sucrose pellets, see *Figure 1*). For clarity, blocks of instrumental performance were considered for analyses on two consecutive sessions (one with the tilt, one with the lever, then averaged for the analysis). Daily training was completed when 30 rewards were earned or 30 min had elapsed. The action-outcome associations and the order of their presentations were counterbalanced between rats and days. For the four first sessions, each action was reinforced. Then, for sessions 5 to 8, a random ratio schedule of 5 was introduced (2 to 10 actions were necessary to obtain the reward, probability of receiving an outcome given a response = 0.2). Sessions 9 to 12 were performed with a RR10 schedule (4 to 20 actions were necessary to obtain the reward, probability of receiving an outcome given a response = 0.1). The last instrumental session with each action (RR10, highlighted in *Figure 1*) was used as a measure of baseline performance for the devaluation test while the last retraining session after this devaluation test (RR10, highlighted in *Figure 1*) was used as a measure of baseline performance for the degradation phase, including the choice test (see *Figure 1*).

## Outcome devaluation test
The day after the last session of training, rats were placed in a plastic feeding cage containing free access of 15 g of one of the two outcomes for one hour of devaluation. Half of the rats in each response-outcome assignment received grain pellets, the remaining receiving sucrose pellets. Immediately after, rats were put in the operant cages for a 10 min extinction test. During the test, both

actions were available but unrewarded. This ensured that rats were using representations of the response-outcome contingencies and outcome value to guide their behavior. Animals that received saline during training also received saline during the test and the same logic applied for animals that had received CNO. Performance was quantified relative to prior baseline levels.

### Consumption test

After the extinction test, rats were put in the plastic feeding cage used for outcome devaluation. They had free access first to 5 g of one outcome for 15 min, and then to 5 g of the other outcome for 15 min. Food consumed was then measured for each outcome. Order of outcome presentation was counterbalanced between rats and groups.

### Degradation procedure

One day after completion of the consumption test, rats received two supplemental sessions of RR10 to reinstate regular instrumental training. Immediately after, the degradation procedure began. For one of the action-outcome associations, the contingency between the action and its consequences was maintained identical to that used during instrumental training (RR10 training) but for the other, the contingency was degraded by delivering the same overall number of rewards randomly even if no action was performed. For both the degradation phase and the test, performance was quantified relative to prior baseline levels.

### CNO preparation and injection

CNO (Enzo Life Science) was diluted in saline with 0.5% of DMSO at a final concentration of 1 mg/ ml. CNO groups received a daily CNO i.p. injection one hour before each training and testing session (1 mg/kg) while saline groups received a daily saline i.p. injection (1 ml/kg of 0.9% Saline) one hour before each training and testing session (see *Figure 1*). We recently demonstrated the efficacy of CNO administration in reducing neuronal activity at a dose of 1 mg/kg using the same reagents and suppliers (*Parkes et al., 2017*). All animals were submitted to surgery then allocated to CNO or saline groups on a random basis prior to training.

### Histology

Rats were perfused transcardially with 150 ml of saline followed by 400 ml of 4% paraformaldehyde (PFA). Brains were kept in the same PFA solution overnight, then sections of 40 μm of the prefrontal cortex and the thalamus were made using a vibratome. Immunochemistry was performed on the sections to enhance the mCherry staining. First, sections were rinsed in PBS 0.1M (5 × 5 min), and then incubated in a blocking solution for 1 hr (4% goat serum and 0.2% Triton X-100 in PBS 0.1 M). Immediately after, sections were put in a bath containing primary antibodies, rabbit anti-RFP (Clinisciences, PM005) primary antibodies diluted at 1/200 in the blocking solution for incubation at 4°C for 48 hr. Sections were then rinsed in PBS 0.1 M (4 × 5 min) and placed for 2 hr in a bath containing a goat anti-rabbit coupled to DyLight 549 (1/200 in PBS 0.1 M) (Jackson ImmunoResearch, 111-025-003) for two hours. Following four 5 min rinses in PBS 0.1 M, Hoechst solution (bisBenzimide H 33258, Sigma, B2883) for counterstaining was added for 15 min (1/5000 in PBS 0.1 M). Finally, sections were rinsed in PB 0.1M (4 × 5 min), mounted in PB 0.05 M onto gelatin-coated slides and coverslipped with the anti-fading reagent Fluoromount G (SouthernBiotech, 0100–01). Images were then captured using a Nanozoomer slide scanner (Hamamatsu Photonics) and analyzed with the NDP.view 2.0 freeware (Hamamatsu Photonics). Histology was performed by FA, VF and MW independently, while being blind to behavioral data.

### Data analysis

The data were submitted to ANOVAs on StatView software (SAS Institute Inc.). For both experiments, Drug (saline/CNO) was the between subject factor, and Devaluation (Devalued/Non Devalued), Degradation (Degraded/Non degraded) and Session (averaged over both actions) were repeated measures when appropriate. The alpha value for rejection of the null hypothesis was 0.05 throughout.

## Acknowledgements

We thank Angélique Faugère and Yoan Salafranque for histological assistance and animal care. This work was supported by an Independent Investigator NARSAD grant #27402 to MW and a grant from the French agency for research ANR-14-CE13-0014 to EC. FA was supported by a BRAIN LabEx PhD extension grant. The microscopy was done in the Bordeaux Imaging Center, a service unit of the CNRS-INSERM and Bordeaux University, member of the national infrastructure France BioImaging, with help from Christel Poujol and Sébastien Marais.

## Additional information

### Funding

| Funder | Grant reference number | Author |
|---|---|---|
| Agence Nationale de la Recherche | ANR-14-CE13-0014 | Etienne Coutureau |
| Brain and Behavior Research Foundation | NARSAD Independent Investigator Grant #24702 | Mathieu Wolff |
| Labex | Brain LABEX PhD Extension Grant 2015 | Fabien Alcaraz |

The funders had no role in study design, data collection and interpretation, or the decision to submit the work for publication.

### Author contributions

Fabien Alcaraz, Validation, Investigation, Methodology; Virginie Fresno, Formal analysis, Validation, Investigation, Visualization; Alain R Marchand, Conceptualization, Software, Formal analysis, Methodology, Writing—review and editing; Eric J Kremer, Conceptualization, Resources, Methodology, Writing—review and editing; Etienne Coutureau, Conceptualization, Supervision, Funding acquisition, Writing—review and editing; Mathieu Wolff, Conceptualization, Supervision, Funding acquisition, Investigation, Visualization, Methodology, Writing—original draft, Project administration, Writing—review and editing

### Author ORCIDs

Mathieu Wolff (iD) http://orcid.org/0000-0003-3037-3038

### Ethics

Animal experimentation: This study was performed in strict accordance with current French (Council directive 2013-118, February 1, 2013) and European (directive 2010-63, September 22, 2010, European Community) laws and policies regarding animal experiments. The experimental protocols received approval #5012053-A from the local Ethics Committee -(C2EA -50, Comité d'éthique pour l'Expérimentation Animale Bordeaux) on December 7, 2012.

### Decision letter and Author response

Decision letter https://doi.org/10.7554/eLife.32517.015
Author response https://doi.org/10.7554/eLife.32517.016

## Additional files

### Supplementary files

• Transparent reporting form
DOI: https://doi.org/10.7554/eLife.32517.007

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

# Appendix 1

DOI: https://doi.org/10.7554/eLife.32517.008

To ensure that CNO injections alone did not alter goal-directed behaviors, three groups of rats (300–330 g at the start of the experiment, obtained from Janvier Labs) were injected with CNO (1 mg/kg; n = 11), saline (0.9%, n = 6) or saline +0.5% DMSO (n = 6). We found that neither CNO nor DMSO affected behavior.

## Instrumental training

Initial instrumental performance was similar to that observed when inhibiting thalamocortical and corticothalamic pathways (*Figure 3A*; *Figure 4A*). As shown in *Appendix 1—figure 1A*, instrumental performance significantly increased (F(5,100) = 146.0, p<0.0001) in a comparable fashion across the three groups. While performance in the saline group appeared to be lower initially, it reached an asymptotical level comparable to that of the other two groups thereafter. Therefore, these analyses revealed no significant effect of Drug (F(2,20) = 2.9, p=0.0773) or of Session X Drug interaction (F(10,100) = 1.2, p=0.3319).

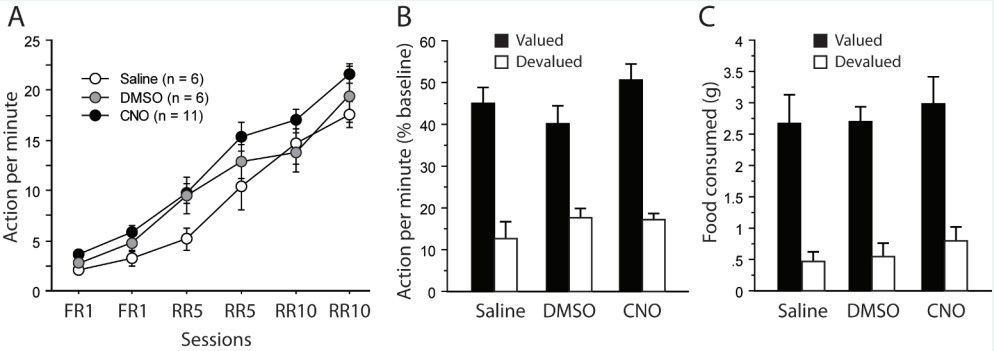

**Appendix 1—figure 1.** Assessment of CNO without DREADD expression. (**A**) Mean number of lever presses (±sem) during instrumental training. (**B**) Mean number of lever presses (±sem, relative to baseline) during the instrumental choice test conducted immediately after selective outcome devaluation under extinction conditions. (**C**) Consumption test.

DOI: https://doi.org/10.7554/eLife.32517.009

## Devaluation test

During the devaluation test that immediately followed sensory-specific satiety, a clear bias towards the action associated for the still valued outcome was evident for all groups (*Appendix 1—figure 1B*). This observation was confirmed by a significant effect of Devaluation (F(1,20) 89.5, p<0.0001). We found no indication that any drug treatment was affecting rats during this test (Drug, F(2,20) = 1.0, p=0.3784; Drug X Devaluation, F(2,20) = 1.5, p=0.2493).

## Consumption test

Immediately after the devaluation test conducted under extinction conditions, a consumption test was performed as described in the methods. All rats expressed a marked preference for the food reward that was not given during satiety (*Appendix 1—figure 1C*). As a result, the main effect of Devaluation was highly significant (F(2,20) = 68.0, p<0.0001). Again, drug treatment produced no detectable effect (Drug, Drug X Devaluation, Fs <1).

## Action-outcome degradation

During the degradation procedure, differential responding was evident for all rats with lower responding when action-outcome contingency was degraded (*Appendix 1—figure 2*). As a

result, the analyses produced a significant effect of Degradation ($F_{(1,20)} = 161.5$, p<0001), Session ($F_{(5,100)} = 2.6$, p=0.0280) and of the Degradation X Session interaction ($F_{(5,100)} = 24.0$, p<0.0001). We found no evidence that any drug treatment would affect performance at this stage (Drug, ($F_{(1,20)} = 1.4$, p=0.2771; Drug X Degradation, Drug X Session, Drug X Degradation X Session, Fs <1).

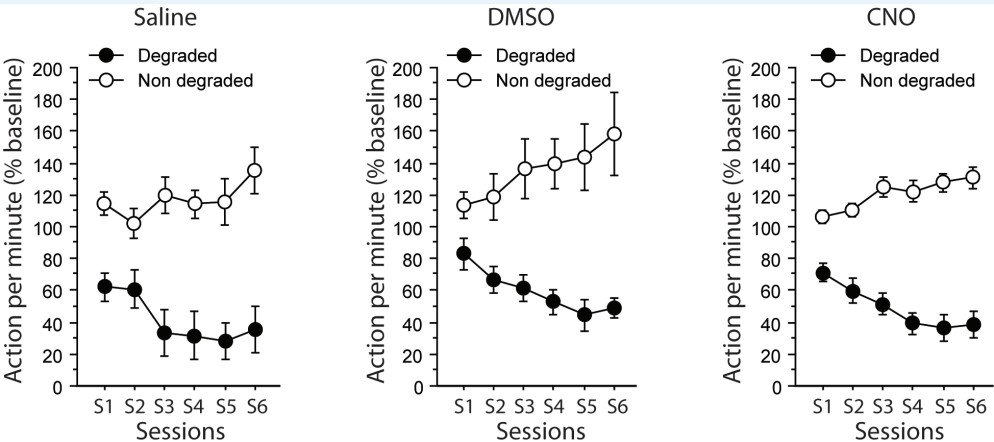

**Appendix 1—figure 2.** Assessment of CNO without DREADD expression. Mean number of lever presses (±sem, relative to baseline) during the contingency degradation procedure.
DOI: https://doi.org/10.7554/eLife.32517.010

## Action-outcome degradation, choice test in extinction:

During the final choice test (*Appendix 1—figure 3*), differential responding was evident for all rats, irrespective of drug treatment. As a result the main effect of Degradation was significant ($F_{(1,20)} = 32.5$, p<0.0001) while the Drug X Degradation interaction was not (F < 1). Responding was slightly higher in the group that received DMSO alone on this occasion, as indicated by the main effect of Drug approaching significance ($F_{(2,20)} = 3.3$, p=0.0589). However the critical CNO *versus* Saline comparison yielded no significant effect (post-hoc Scheffe's test, p=0.6381). In addition, further analyses conducted on each group separately confirmed the existence of a Degradation effect for the DMSO-treated ($F_{(1,5)} = 8.5$, p=0.0334) and the CNO-treated ($F_{(1,10)} = 23.1$, p=0.0007) groups while it approached significance for the saline group ($F_{(1,5)} = 6.0$, p=0.0579).

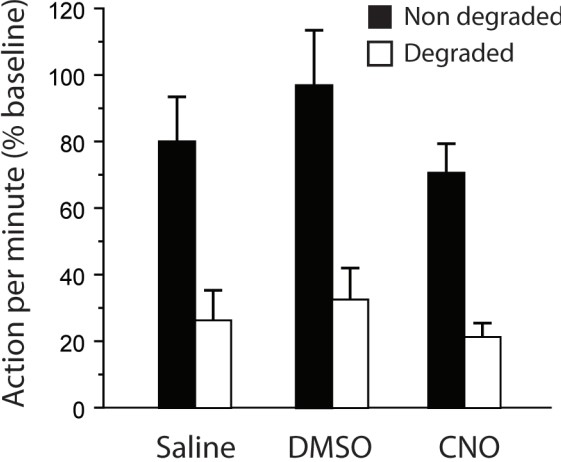

**Appendix 1—figure 3.** Assessment of CNO without DREADD expression. Mean number of lever presses (±sem, relative to baseline) for the final choice test conducted under extinction conditions.
DOI: https://doi.org/10.7554/eLife.32517.011

Thus, altogether, these supplemental data confirmed that CNO treatment alone could not account for the effects observed in the main study.

