## [Decision Letter]

Thank you for submitting your article "Thalamocortical and corticothalamic pathways support distinct goal attributes" for consideration by *eLife*. Your article has been favorably evaluated by Timothy Behrens (Senior Editor) and three reviewers, one of whom, Geoffrey Schoenbaum (Reviewer #1), is a member of our Board of Reviewing Editors. The following individuals involved in review of your submission have agreed to reveal their identity: Sean Ostlund (Reviewer #2); Laura Corbit (Reviewer #3).

The reviewers have discussed the reviews with one another and the Reviewing Editor has drafted this decision to help you prepare a revised submission.

Summary:

The authors use genetic techniques to test the hypothesis that reciprocal connections between PFC and MD thalamus might play distinct roles in goal directed behavior. They report that projections to MD are critical for selective changes in instrumental behavior after devaluation, whereas projections from MD are necessary for selective changes in instrumental behavior after both devaluation and continency degradation.

Essential revisions:

There are two essential revisions. The first is the need for a control group to rule out CNO effects, given the recent report by Michaelides that CNO has its effect through peripheral conversion to clozapine, which then enters the CNS and could act at other sites. The addition of a separate group and comparison to the controls might be acceptable, rather than repeating the entire experiment, to rule this out.

The second essential revision is to address the alternative interpretations given the effect of CNO/DREADD on response rates in the critical tests. A fair discussion of this or additional data to rule out extinction, generalization and a floor effect on responding is necessary.

*Reviewer #1:*

In this study, the authors use genetic approaches to explore the effects of specifically inactivating connections between DMPFC and MD thalamus on instrumental devaluation and contingency degradation. They are exploring the interesting and novel question of whether projections in different directions between these areas may play dissociable roles in the behavior. They use DREADD receptors and retrogradely transported CRE recombinase to target directional projections and inject CNO systemically during instrumental learning, devaluation and contingency degradation testing to inactivate the projections. They show that projections to MD are critical for selective changes in instrumental behavior after devaluation, whereas projections from MD are necessary for selective changes in instrumental behavior after both devaluation and continency degradation. From this they conclude that the former projection is necessary for current value representations and the latter for maintaining or using action-outcome representations. At least that is my reading.

The study is well-conceived and tests an interesting and important question. In my opinion, it has two major problems however. One is not specific to this study, but is critical nonetheless, and it involves the recent report that the DREADD receptors are activated by clozapine and not by peripherally administered CNO. Specifically, my understanding of these data is that they show that CNO does not in fact enter the brain when injected peripherally. Instead it is converted to clozapine, and it is clozapine that acts on the receptor. Obviously, clozapine also has other endogenous sites of action. This means that the two-group approach used here is not sufficient for using this tool, since any effect in the experimental group could be due to the interaction between the clozapine and the DREADD receptor, as planned by the authors, or it could reflect the action of clozapine in some other area. To rule out the latter interpretation requires a CNO only control group. There is really no way around this it seems to me, given this recent report. Given that the authors were presumably blindsided by this new finding, I think the addition of this group in supplemental or in some other way would be acceptable (i.e. it is not necessary to repeat the entire experiment), but I think it must be added, otherwise we all look foolish.

The second problem I have is specific to this experiment, and it is the low levels of responding in the experimental groups. That is, the critical effects occur largely because the CNO/DREADD group stops responding. It seems to me that this could reflect the functions the authors claim. However it could also reflect a simple lack of responding, or it could also reflect more rapid extinction learning or generalization I think, independent of the claimed effects. This problem is not unique to this experiment; many studies like this suffer from this issue. It also affects both experiments, but perhaps is more of a problem in the first. I think it is important to note it clearly in the discussion of the effects, along with whatever mitigating observations the authors want to make. To directly address I think one might restrict the analysis of devaluation and contingency degradation to performance matched pairs. That is, the authors could analyze rats that responded similarly at the end of training, or that responded similarly to the valued or non-degraded response in the test. If the impact of treatment on the devaluation or degradation persisted in these specific rats, this would be good evidence that it was independent of any overall performance effect. Or one might look at responding across time in the critical test sessions to rule out a general increase in extinction in the face of non-reward. In any event, I think this must be fairly discussed. Maybe one of these other functions is really what is going on?

*Reviewer #2:*

This study investigates an important topic using techniques that are generally appropriate. The findings tend to support the authors' claim that direct connections between the dmPFC and MD thalamus are important for goal-directed behavior, with dmPFCMD and MDdmPFC projections underlying sensitivity to outcome value but only MDdmPFC underlying action-outcome contingency. Although these findings would be novel, there are a few issues that make data interpretation difficult, and that limit the significance of this project, as it stands.

1) Since no DREADD-free controls are used, it is unclear if CNO's effects on behavior are dependent on hM4Di expression. Given recent concerns about the back conversion of CNO to clozapine, a drug which has actions at various endogenous receptors, it is highly recommend that all DREADD studies include such controls. The authors discuss this issue and offer an argument that CNO did not disrupt certain behaviors (e.g., consumption) and had partially dissociable effects in groups in which different pathways were targeted. The fact that outcome devaluation was disrupted by CNO in both groups, however, leaves open the possibility that this aspect of behavior is disrupted by unconditional CNO effects. It therefore seems prudent to add additional data that speaks to this alternative account. Also, although the concentration of CNO is described, it's unclear what dose was used for treatment and whether it was consistently used across training/testing stages. Further discussion of how this dose was selected and how repeated exposure to CNO might be expected to impact the efficacy of this treatment would also be appropriate. Finally, it seems that saline was used as a control treatment rather than the actual DMSO-containing vehicle. Please clarify. This makes it that much more important to assess the unconditional effects of the CNO-DMSO solution.

2) In both experiments the CNO group showed generally low levels of responding during devaluation testing (rather than indiscriminate responding). This is not surprising given the results of lesions studies, but the authors should consider that a floor effect obscured detection of sensitivity to devaluation. Importantly, test data are presented as% baseline performance. Therefore this is a bigger concern for experiment 1 (MD-dmPFC group) since CNO had a substantial effect on baseline (training) performance, and also seems to have further suppressed responding at test to very low levels. (Indeed, despite this and the relatively small n's, there was a nonsignificant trend towards devaluation (p = 0.15) in this group). I would be interested to know whether similar deficits were observed early in test sessions, when response rates were likely higher. It is important to note that although the authors explain that the CNO did not differ from the vehicle group at the end of training, the rest of the training data suggests otherwise. There was most likely insufficient power to find significant effects on any given training day (despite the clear trend).

3) The methods do not clearly state how CNO was delivered during contingency training, e.g., that the same rats get CNO during both rounds of training/testing. More should be done clarify the CNO treatment during retraining sessions. Also, apparently there was an initial devaluation test in which all rats were tested on vehicle. This data was not presented but would speak to whether chemoinhibition of targeted pathways during encoding only (not at test) were sufficient to disrupt action-outcome based response retrieval. In general, discussion of literature indicating the stage-limited roles for dmPFC and MD in goal-directed learning but not performance is somewhat vague (e.g., Results, first paragraph and Discussion, fourth paragraph) and may be confusing to the reader, particularly given that the main approach and findings here do not attempt to investigate this further.

*Reviewer #3:*

This article by Alcaraz and colleagues uses a chemogenetic approach (DREADDs) to investigate the specific role of thalamocortical and corticothalamic pathways in instrumental learning and control. Both the medial prefrontal cortex (particularly the prelimbic region) and the mediodorsal thalamus have been shown to be involved in instrumental learning in previous work using lesion methods but these previous methods do not allow the role of the individual circuits to be isolated. The current results are an important contribution and advancement in that regard.

The manuscript uses appropriate methods for both behavioural and neural manipulations allowing strong conclusions to be drawn from the data. These findings make an important contribution to understanding of the circuits underlying instrumental learning and corticothalamic interactions in general. Further, they highlight the need to carefully examine the role of each pathway within reciprocal circuits to fully appreciate their contribution to behavior.

I have several specific comments for the authors to consider.

- I thought a schematic outlining the design, perhaps added as a panel in each figure, would greatly aid understanding of the behavioural design without going into great detail in the body of the manuscript (when was CNO given (training, testing, retraining, etc.?), point out that animals are trained on two R-O relationships (R1-O1, R2-O2 etc.)).

- The sample size ends up being rather small after exclusions. While I have a difficult time imagining how adding more animals would change the overall pattern of results, it would allow stronger conclusions and assuage any reservations in readers less familiar with these methods. For example, in Experiment 1, it's awkward that the groupxdevaluation interaction is not significant. I can live with this as the simple effects confirm the impairment in the CNO group and not in the saline group which is very much what's suggested by the data in Figure 1. But the authors are then forced to describe the impairment as mild, thereafter. To me, this has implications for interpretation; is the impairment weak or is the power weak? While subtle, different conclusions might be reached if power could be ruled out as a contributor to the effect. So I'm not requiring more experiments, and perhaps the authors are attempting to tread lightly considering their statistical support, however, it would be worth giving further consideration to the description of the results.

- I felt the histology figures were too small for the reader to make their own assessment of the expression of virus and it wasn't possible to evaluate the extent of overlap of the two vectors. The description of the histology methods and analyses was very brief. I am confident that placements were in the targeted area but there may be more subtle aspects of expression (degree, things like cortical layers, etc.) that readers may like to see themselves. Could the image be larger or a zoom be inset?

- Subsection “CNO preparation and injection” – here there is peculiar mention of all groups receiving saline before "the first session of tests" and then being split into saline and CNO groups for a section devaluation test. Is this what happened? I thought rats were in consistent treatment groups throughout and only data for a single devaluation test is reported? Please clarify. This isn't entirely trivial as there's some indication in the literature that the MD may contribute to acquisition but not expression of R-O learning.

---

## [Author Response]

Reviewer #1:[…] The study is well-conceived and tests an interesting and important question. In my opinion, it has two major problems however. One is not specific to this study, but is critical nonetheless, and it involves the recent report that the DREADD receptors are activated by clozapine and not by peripherally administered CNO. Specifically, my understanding of these data is that they show that CNO does not in fact enter the brain when injected peripherally. Instead it is converted to clozapine, and it is clozapine that acts on the receptor. Obviously, clozapine also has other endogenous sites of action. This means that the two-group approach used here is not sufficient for using this tool, since any effect in the experimental group could be due to the interaction between the clozapine and the DREADD receptor, as planned by the authors, or it could reflect the action of clozapine in some other area. To rule out the latter interpretation requires a CNO only control group. There is really no way around this it seems to me, given this recent report. Given that the authors were presumably blindsided by this new finding, I think the addition of this group in supplemental or in some other way would be acceptable (i.e. it is not necessary to repeat the entire experiment), but I think it must be added, otherwise we all look foolish.

We have added the required controls, which are included as supplemental material. We repeated the whole experimental procedure (depicted in the new Figure 1) with three new and independent groups that were treated with either 1 mg/kg CNO (n = 11), 0.9% saline (n = 6) or saline + DMSO (n = 6) to establish this important control (the latter was a suggestion from reviewer 2). These new data show that CNO injections alone, in the absence of DREADDs, did not affect behavior at any stage of testing.

The second problem I have is specific to this experiment, and it is the low levels of responding in the experimental groups. That is, the critical effects occur largely because the CNO/DREADD group stops responding. It seems to me that this could reflect the functions the authors claim. However it could also reflect a simple lack of responding, or it could also reflect more rapid extinction learning or generalization I think, independent of the claimed effects. This problem is not unique to this experiment; many studies like this suffer from this issue. It also affects both experiments, but perhaps is more of a problem in the first. I think it is important to note it clearly in the discussion of the effects, along with whatever mitigating observations the authors want to make. To directly address I think one might restrict the analysis of devaluation and contingency degradation to performance matched pairs. That is, the authors could analyze rats that responded similarly at the end of training, or that responded similarly to the valued or non-degraded response in the test. If the impact of treatment on the devaluation or degradation persisted in these specific rats, this would be good evidence that it was independent of any overall performance effect. Or one might look at responding across time in the critical test sessions to rule out a general increase in extinction in the face of non-reward. In any event, I think this must be fairly discussed. Maybe one of these other functions is really what is going on?

Responding across time:

We have conducted supplemental analyses and produced a new paragraph in the Discussion dealing with these issues (fifth paragraph). Analyzing the rate of responding over time during the devaluation test (treated as five 2 min blocks) confirmed the existence of extinction with a significant effect of Block, which did not interact with Drug (Experiment 1: Block (F(9,108) = 8.3, P < 0.0001); Block X Drug (F(9,108) = 1.2, P = 0.2908); Block X Devaluation X Drug (F(9,108) = 1.6, P = 0.1335); Expt2: Block (F(19,108) = 3.9, P = 0.0002); Block X Drug (F(9,108) = 1.3, P = 0.2721); Block X Devaluation X Drug (F(9,108) = 1.4, P = 0.2062). Moreover, analyses conducted separately on saline- and CNO-treated rats confirmed that responding gradually decreased over time during this test in each group (Saline: Block, F(4,24) = 6.6, P = 0.0010; CNO: Block, F(4,24) = 7.4, P = 0.0005), see the figures produced below). These data are now included in the corresponding Results sections. We found no evidence supporting the view that extinction processes during that test could be differently affected by saline versus CNO treatment.

Floor effect:

The same analysis allows us to also address the issue of a potential performance floor effect (a concern also raised by reviewer 2). It provides evidence showing that performance was not minimal at the start of the devaluation test for both groups of animals, especially when inhibiting thalamocortical pathway. Indeed, CNO-treated rats responded initially at higher rates for the action associated with the devalued outcome as shown by Author response image 1 and the related analyses produced below.

**Author response image 1. respfig1:** Experiment 1: inhibiting thalamocortical pathway.

When focusing only on the initial 2 min block during which responding was maximal, the analyses revealed an overall effect of Devaluation (F(1,12) = 4.8, P = 0.0499) but not of Drug (F<1). Interestingly, the critical Drug X Devaluation now reached significance (F(1,12) 4.7, P = 0.0501). Further analyses aimed at comparing responding rates for actions associated with the valued and the devalued outcome during this initial 2 min block produced the following: while the main effect of Drug did not reach significance for the still valued option (F(1,12) = 2.5, P = 0.1338), it approached significance for the devalued one (F(1,12) = 3.7, P = 0.0782), CNO-treated rats tended to respond *more* than saline-treated rats). Thus, the specific impairment exhibited by CNO-treated rats was also evident when responding was maximal and therefore, cannot result directly from performance floor effect.

Qualitatively similar findings were evident when considering initial responding (i.e. during the initial 2 min block) during the devaluation test performed for experiment 2 (inhibiting corticothalamic pathway). Both the effect of Devaluation and Drug reached significance (F(1,12) = 7.8, P = 0.0164; F(1,12) = 6.3, P = 0.0271, respectively), as did the Drug X Devaluation interaction (F(1,12) = 10.7, P = 0.0066). On this instance responding was lower in the CNO-treated group for the still valued option (F(1,12) = 10,1 P = 0.0079) but not for the devalued one (F<1).

On top of these additional data, we acknowledge that overall responding was low during devaluation tests. Overall, levels of performance seem to vary depending on labs and batches of rats. For example, in this recent study from Balleine and colleagues, as little as 3 (Devalued) or 5 (Non-devalued) presses per minute were observed during an initial 10 min devaluation test (Bradfield and Balleine, JN2017, Figure 1, even lower values are reported thereafter) following instrumental training with a RR20 schedule, instead of a RR10 in the present study (which should result in even higher levels of responding). Similarly, 1 (Dev) vs. 8 (NDev) for Sham or 2 (Dev) vs. 2 (NDev) presses / minute for the MD group were reported in the classic MD lesion study from Corbit et al., EJN 2003. These comments have been added in the Discussion (fifth paragraph). Please also note that performance during the degradation procedure was not affected by CNO treatment during training, but specifically impaired during the test for the TC group only, with substantial levels of performance on this instance.

Generalization:

Finally, while we cannot exclude a possible generalization, we used two clearly distinct manipulanda (a lever and a tilt), which should limit any generalization as now more clearly pointed in the Discussion (fifth paragraph). It is possible that what appears to be generalization may actually result from an inability to select the correct option in the absence of the sensory feedback provided by reward to guide behavior.

Reviewer #2:This study investigates an important topic using techniques that are generally appropriate. The findings tend to support the authors' claim that direct connections between the dmPFC and MD thalamus are important for goal-directed behavior, with dmPFCMD and MDdmPFC projections underlying sensitivity to outcome value but only MDdmPFC underlying action-outcome contingency. Although these findings would be novel, there are a few issues that make data interpretation difficult, and that limit the significance of this project, as it stands.1) Since no DREADD-free controls are used, it is unclear if CNO's effects on behavior are dependent on hM4Di expression. Given recent concerns about the back conversion of CNO to clozapine, a drug which has actions at various endogenous receptors, it is highly recommend that all DREADD studies include such controls. The authors discuss this issue and offer an argument that CNO did not disrupt certain behaviors (e.g., consumption) and had partially dissociable effects in groups in which different pathways were targeted. The fact that outcome devaluation was disrupted by CNO in both groups, however, leaves open the possibility that this aspect of behavior is disrupted by unconditional CNO effects. It therefore seems prudent to add additional data that speaks to this alternative account. Also, although the concentration of CNO is described, it's unclear what dose was used for treatment and whether it was consistently used across training/testing stages. Further discussion of how this dose was selected and how repeated exposure to CNO might be expected to impact the efficacy of this treatment would also be appropriate. Finally, it seems that saline was used as a control treatment rather than the actual DMSO-containing vehicle. Please clarify. This makes it that much more important to assess the unconditional effects of the CNO-DMSO solution.

We have clarified that the CNO dose was 1 mg/kg as this dose is highly standard and was previously shown to be sufficient to elicit neuronal inhibition in vitro, in a recently published study from our group relying on similar reagents and procedures (Parkes et al., 2017). See in particular subsection “CNO preparation and injection” in the Materials and methods.

We have produced the requested controls as supplemental material. These show that neither treatment with 1 mg/kg of CNO, nor DMSO administration alone alter behavior at any stage of testing.

2) In both experiments the CNO group showed generally low levels of responding during devaluation testing (rather than indiscriminate responding). This is not surprising given the results of lesions studies, but the authors should consider that a floor effect obscured detection of sensitivity to devaluation. Importantly, test data are presented as% baseline performance. Therefore this is a bigger concern for experiment 1 (MD-dmPFC group) since CNO had a substantial effect on baseline (training) performance, and also seems to have further suppressed responding at test to very low levels. (Indeed, despite this and the relatively small n's, there was a nonsignificant trend towards devaluation (p = 0.15) in this group). I would be interested to know whether similar deficits were observed early in test sessions, when response rates were likely higher. It is important to note that although the authors explain that the CNO did not differ from the vehicle group at the end of training, the rest of the training data suggests otherwise. There was most likely insufficient power to find significant effects on any given training day (despite the clear trend).

The reviewer is right in pointing the similarity between MD chemogenetic inhibition with that of neurotoxic MD lesions on levels of responding (e.g. Corbit et al., 2003). This observation is now mentioned in the Discussion (fifth paragraph). Please note however that the same animals did respond at a rate comparable with that of controls during the degradation phase.

Furthermore, we have provided extended supplemental analyses of devaluation tests above to address this point (see reviewer 1). As a short reminder, when focusing only on the initial 2 min block during the devaluation test, during which responding was maximal (see Author response image 1 and Author response image 2), the analyses yielded qualitatively similar effects. Interestingly, for experiment 1 (inhibiting TC pathway), that analysis revealed an overall effect of Devaluation (F(1,12) = 4.8, P = 0.0499) but not Drug (F<1), but the critical Drug X Devaluation now reached significance (F(1,12) 4.7, P = 0.0501). For experiment 2 (inhibiting CT pathway), the results were largely consistent with those obtained when analyzing the whole test: both the effect of Devaluation and Drug reached significance (F(1,12) = 7.8, P = 0.0164; F(1,12) = 6.3, P = 0.0271, respectively), as did the Drug X Devaluation interaction (F(1,12) = 10.7, P = 0.0066).

**Author response image 2. respfig2:** Experiment 2: inhibiting corticothalamic pathway.

Thus, the major findings reported in the manuscript are confirmed by restricted analyses on the beginning of the test and supplemental analyses on the dynamics of responding during devaluation test have been added in the Results as they are sufficient to rule out any potential effect of chemogenetic treatment on extinction, or on floor effect alone as the main driver of the impairment.

3) The methods do not clearly state how CNO was delivered during contingency training, e.g., that the same rats get CNO during both rounds of training/testing. More should be done clarify the CNO treatment during retraining sessions. Also, apparently there was an initial devaluation test in which all rats were tested on vehicle. This data was not presented but would speak to whether chemoinhibition of targeted pathways during encoding only (not at test) were sufficient to disrupt action-outcome based response retrieval. In general, discussion of literature indicating the stage-limited roles for dmPFC and MD in goal-directed learning but not performance is somewhat vague (e.g., Results, first paragraph and Discussion, fourth paragraph) and may be confusing to the reader, particularly given that the main approach and findings here do not attempt to investigate this further.

Methods have been clarified with respect to the CNO administration, see in particular the new Figure 1, stating clearly that CNO was administered before each behavioral session (except magazine training), which included retraining.

We apologize that the corresponding paragraph of the Materials and methods regarding CNO administration was misleading. It has been re-written as no initial test under vehicle was conducted. In addition, we added in the conclusion a fair mention to the data available regarding stage-limited roles for both the dmPFC and the MD and we now suggest that the present work could be expanded by performing similar interventions at specific stages of the task.

Reviewer #3:[…] I have several specific comments for the authors to consider.- I thought a schematic outlining the design, perhaps added as a panel in each figure, would greatly aid understanding of the behavioural design without going into great detail in the body of the manuscript (when was CNO given (training, testing, retraining, etc.?), point out that animals are trained on two R-O relationships (R1-O1, R2-O2 etc.)).

This has been done (new Figure 1).

- The sample size ends up being rather small after exclusions. While I have a difficult time imagining how adding more animals would change the overall pattern of results, it would allow stronger conclusions and assuage any reservations in readers less familiar with these methods. For example, in Experiment 1, it's awkward that the groupxdevaluation interaction is not significant. I can live with this as the simple effects confirm the impairment in the CNO group and not in the saline group which is very much what's suggested by the data in Figure 1. But the authors are then forced to describe the impairment as mild, thereafter. To me, this has implications for interpretation; is the impairment weak or is the power weak? While subtle, different conclusions might be reached if power could be ruled out as a contributor to the effect. So I'm not requiring more experiments, and perhaps the authors are attempting to tread lightly considering their statistical support, however, it would be worth giving further consideration to the description of the results.

Inhibiting the MD-to-dmPFC pathway during devaluation test produced mixed findings, due to the lack of interaction. See however the additional analyses showing clear evidence when focusing only on the beginning of the test (responses to reviewer 1 and 2). In addition, the impairment produced by inhibiting the dmPFC-to-MD pathway on the same occasion is very clear (with comparable group size) and inhibiting the TC pathway during degradation also produced a clear impairment. With these observations in mind, together with the extended analyses produced above, we are confident that the impairment exhibited by CNO-treated rats during experiment 1 is mild and does not artificially derive from weak power per se.

- I felt the histology figures were too small for the reader to make their own assessment of the expression of virus and it wasn't possible to evaluate the extent of overlap of the two vectors. The description of the histology methods and analyses was very brief. I am confident that placements were in the targeted area but there may be more subtle aspects of expression (degree, things like cortical layers, etc.) that readers may like to see themselves. Could the image be larger or a zoom be inset?

We have added a new Figure 2 featuring additional magnifications of mCherry expression at the level of both the dmPFC and the MD. The description of the related histology has been substantially expanded.

- Subsection “CNO preparation and injection” – here there is peculiar mention of all groups receiving saline before "the first session of tests" and then being split into saline and CNO groups for a section devaluation test. Is this what happened? I thought rats were in consistent treatment groups throughout and only data for a single devaluation test is reported? Please clarify. This isn't entirely trivial as there's some indication in the literature that the MD may contribute to acquisition but not expression of R-O learning.

As noted by reviewer 2, this is the result of the initial formulation, which was not appropriate and unfortunately misleading. The paragraph has been corrected since there was no such test.